# Secondary Metabolites in Nectar-Mediated Plant-Pollinator Relationships

**DOI:** 10.3390/plants12030550

**Published:** 2023-01-25

**Authors:** Marta Barberis, Daniele Calabrese, Marta Galloni, Massimo Nepi

**Affiliations:** 1Department of Biological, Geological and Environmental Sciences, University of Bologna, Via Irnerio 42, 40126 Bologna, Italy; 2Department of Life Sciences, University of Siena, Via P.A. Mattioli 4, 53100 Siena, Italy; 3National Biodiversity Future Centre (NBFC), 90123 Palermo, Italy

**Keywords:** floral nectar, secondary compounds, plant-pollinator-microbe interactions, pollinator behaviour

## Abstract

In recent years, our understanding of the complex chemistry of floral nectar and its ecological implications for plant-pollinator relationships has certainly increased. Nectar is no longer considered merely a reward for pollinators but rather a plant interface for complex interactions with insects and other organisms. A particular class of compounds, i.e., nectar secondary compounds (NSCs), has contributed to this new perspective, framing nectar in a more comprehensive ecological context. The aim of this review is to draft an overview of our current knowledge of NSCs, including emerging aspects such as non-protein amino acids and biogenic amines, whose presence in nectar was highlighted quite recently. After considering the implications of the different classes of NSCs in the pollination scenario, we discuss hypotheses regarding the evolution of such complex nectar profiles and provide cues for future research on plant-pollinator relationships.

## 1. Introduction

Pollination by insects is an ecosystem service that maintains planetary biodiversity and ecosystem functions. It is also fundamental for human food security. About 90% of the currently known angiosperm species, totalling just under 300,000 species [1], are pollinated by insects, and more than 1500 crops around the world benefit from the same services [2]. Pollen and nectar are the primary alimentary rewards offered by plants to floral visitors, and of the two, nectar is sought by a wider range of animals, mediating the majority of plant-animal relationships [3]. Nectar is a concentrated sugary secretion containing a combination of simple sugars (sucrose, glucose, and fructose) [4]. This ready-to-use energy source powers the flight of feeding insects, birds, and other animals [4,5,6]. A co-evolutionary relationship between the relative percentage of sugar in nectar and the food preferences of pollinators was revealed in the early 1980s [7]. Although nectar amino acids occur at much lower concentrations than sugars, they are a source of nitrogen for pollinators and contribute to the taste of nectar [4,5,6]. All 20 protein-building amino acids have been detected in nectar [8] and reference therein], and insect preferences for specific amino acids are also known [9,10].

For decades, nectar chemistry studies concerned the analysis of sugars and amino acids, focusing on their basic importance as food rewards in the framework of the mutualistic relationship between plants and pollinators. This classical view of floral nectar was recently challenged by studies focusing on substances present at low concentrations in nectar and not directly related to its food value, i.e., nectar secondary compounds (NSCs) [11,12]. Since several secondary compounds in plants are known to deter herbivores and to have antimicrobial properties [13], NSCs were initially thought to defend against opportunistic nectar-feeding animals and nectar-dwelling microorganisms, protecting plants from the exploitation of their nectar ([11] and references therein). The former case was formalised as the “nectar forager selection” hypothesis, where opportunistic nectar-feeding animals were identified as scarcely efficient pollinators or nectar thieves/robbers [14]. The latter case was instead formalised as the “antimicrobial” hypothesis [15]. A series of studies indeed confirmed the functions suggested in such hypotheses e.g., [16,17,18], but at the same time other studies clarified that NSCs do not solely play these roles: it became clear, in fact, that they can affect insect foraging behaviour in several additional ways [19,20,21], with potential effects on pollination efficiency and plant reproductive success. In this regard, it is interesting to note that NSC concentrations are often lower than those found in plant tissues, where secondary compounds have a clear deterrent effect on herbivorous insects e.g., [22]. Since the effect of secondary compounds on insects is dose-dependent e.g., [22,23], it is plausible that NSCs may have functions other than deterrence.

It has since been highlighted that some NSCs affect an array of insect behavioural traits of particular interest in the scenario of foraging activity and pollination of flowers: phagostimulation [13,24], locomotion [20,25], learning and memory [19,26,27], arousal and aggressiveness [28], olfactory perception [29], phototaxis [30], reward-seeking [21], and social communication [31,32]. According to the recent “manipulation” hypothesis, NSCs can be regarded as tools available to plants for manipulating the behaviour of foraging insects and exploiting their mutualistic interactions: plants rewarding pollinators with “doped” nectar maximise the benefits they obtain, increasing the efficiency of the pollination service [33]. Although this hypothesis has some gaps (e.g., lack of experimental evidence directly linking NSCs, pollination efficiency, and plant fitness), it opens new ecological and evolutionary scenarios. Here, we bring together the actual knowledge on the plethora of roles played by the most important classes of nectar compounds, with a particular focus on the recently discovered class of biogenic amines, whose presence in floral nectar raises a series of interesting new questions.

## 2. Nectar Phenols

Phenols are organic compounds with one or more six-carbon aromatic rings carrying one or more hydroxy groups [34]. They are quite common in floral nectar [8,15,35,36]: indeed, more than 30% of plant species seem to secrete phenolic nectar [35]. Their ecological role, as well as that of other NSCs, was initially assumed to be a deterrent to scarcely efficient pollinators [8] and nectar thieves such as ants [37]. Interestingly, when it was confirmed that phenols in nectar can deter undesirable visitors [32,38,39,40], it was simultaneously found that they can attract effective pollinators, reinforcing pollinator fidelity to the plant [41]. The study conducted by Gong et al. [29] provides an interesting example of how nectar polyphenols rule complex interactions beyond the simple deterrence/attraction dichotomy: the results demonstrate that honeybees show a preference for solutions containing polyphenols and that these compounds are capable of increasing memory retention and affecting sensitivity to bee-alarm odours. These alarm odours are pheromones that insects can emit while feeding on flowers to alert nest mates to danger [42]. If polyphenols increase bee sensitivity to such odours, then the visitation rate of bees to flowers marked with such pheromones may decrease. This suggests a negative impact of nectar polyphenols on plant fitness, possibly determining reduced pollination and seed set. Nevertheless, a second scenario is also possible: if there are few sources of danger, the number of flowers marked with alarm odour is low, and increased sensitivity to such signals may reduce visits to flowers that have already been visited, favouring not yet visited flowers.

Nevertheless, we are still discovering actions that phenols seem to exert in floral nectar: for example, they seem to be feeding stimulants for some insects [43], while others have antibacterial and antifungal properties [44,45,46]. With reference to the latter function, strong antifungal and antibacterial activities of plant tannins have been confirmed [47,48]. These tannins are natural water-soluble polyphenols of variable molar mass [49], often detected in floral nectar [50]. Their antimicrobial function is important since it may reduce the proliferation of nectar-dwelling fungi and bacteria, commonly found in nectar, which deplete the food value of nectar by exploiting sugars and amino acids for their own metabolism [51,52,53].

Other nectar phenols are responsible for coloured nectar, which most authors consider to be an honest signal for floral visitors [54,55]. The dark colour of some nectar can be due to the oxidation of phenolic compounds, and the colour is generally lighter in young flowers [54] (e.g., in Figure 1, personal observation). Coloured nectar can facilitate remote detection of a food source by pollinators, as well as provide an assessment of nectar quantity in individual flowers e.g., [56,57]. However, there are other possible explanations for coloured nectar, such as being a deterrent to nectar thieves or having an anti-microbial effect that preserves the quality of the food resource in long-lasting flowers. Neither explanation is mutually exclusive [54]. For instance, the dark purple nectar of *Leucosceptrum canum* is due to the anthocyanidin 5-hydroxyflavylium, the role of which may go beyond that of a simple attractant. Birds visiting the flowers of *L. canum* are reported to feed only when the nectar becomes palatable, which coincides with the reproductive maturity of the flower and increases pollination efficiency while protecting immature flowers from damage or nectar depletion [55]. Such bird behaviour may be driven by the process of the oxidation of the compound, which is known to be highly unstable.

Vividly coloured nectars are found in few plant species and are considered a rare floral trait [54]. Some phenols have even fluorescent properties e.g., [58], but our understanding of the phenomenon is still limited. The ecological meaning of fluorescent nectars has been suggested to be guiding pollinators that see in the UVA band towards the flowers, however, not all authors agree on the veracity of this hypothesis [59,60].

Even though the majority of species present scentless nectar, another interesting phenomenon worth to be mentioned and involving phenols (and terpenoids, see next section) is that of scented nectars [61]. Scented compounds may be dissolved in the aqueous medium of nectar and absorbed passively from the surrounding floral tissues [4,62]. Since floral scents are heterogeneous bouquets of chemicals [63], it is easy to imagine that scented nectars are likewise a complex combination of compounds and not mere attractants. They likely have antimicrobial activity [64,65], play a role in defence physiology, or act as signals to predators and parasitoids [66].

An interesting case concerns plant scents and mate location by pollinators. Mate location often involves species-specific insect pheromones, which have long been considered a major factor for mate-finding success [67]. However, Xu and Turlings [68] suggested that plant volatiles play a crucial role as coadjutants in insect reproduction: pollinators are often stimulated to release more pheromones and/or increase mate receptivity by plant volatiles. Although the authors studied volatiles released from various plant tissues (e.g., leaves, flowers, fruits), it is reasonable to transpose this further ecological role also to nectar scents, which in most cases originate from the volatiles of the surrounding tissues e.g., [61].

The study by Raguso [61] confirms that in some of the species presenting scented nectar, nectar odours are like those emitted by floral tissues, but intriguingly, the pattern of nectar sharing similar chemical scents with floral tissues is not confirmed for other species, the nectar of which shows a unique bouquet of chemicals.

Along with all the possible functions listed so far, it is also worth mentioning that some plants produce hallucinogenic or narcotic substances that affect pollinator behaviour, disorienting their flight which is often described as sluggish or drunken [69]. This seems to be determined by phenol derivatives [70] or alkaloids [71,72], and appears–at first glance–like a counter-intuitive effect. Whether these substances create addiction or whether floral visitors may find the effect of “getting high” rewarding in itself (things that would both enhance their fidelity) remains to be clarified. In any case, a possible ecological explanation for the presence and maintenance of such compounds in floral nectar could be that sluggish behaviour prolongs the time spent by visitors on the flower, increasing the chance of pollination.

## 3. Nectar Terpenoids

Terpenoids are a large and diverse class of naturally occurring compounds derived from five carbon isoprene units, differentiated from each other by their basic skeleton and functional groups [34]. They are the main constituents of essential oils and have been detected in the floral nectars of a good number of plant species [61,73,74]. Although terpenoids are generally thought to be insect attractants [75,76,77], Junker and Blüthgen [78] confirmed a repellent effect of specific terpenoids commonly found in floral scents, suggesting that their presence in floral nectar may discourage nectar thieves or protect against fungal diseases [79]. Interestingly, many terpenoids also produce satiety in insects [80].

The case of the nectar terpenoid triptolide, which is found in the floral nectar of *Tripterygium hypoglaucum*, highlights that certain secondary metabolites are tolerated differently by closely related insect taxa. Triptolide is known to impair honeybee foraging responses, dance communication, and olfactory learning [81]. This specific example supports a coevolution hypothesis since the sympatric species *A. cerana* shows higher tolerance to the toxin than the introduced species *A. mellifera* [81].

Another important role of nectar terpenoids (and alkaloids, see next section), is to enhance insect immune response to parasites and promote floral-visitor health. The nectar terpenoid abscisic acid, for instance, improves the immune response of worker honeybees and larvae attacked by *Varroa destructor* [82], while both classes of chemicals significantly reduce the load of the intestinal parasite *Crithidia bombi* in bumblebee colonies, playing a crucial role in controlling transmission within and between colonies [83]. Since a mechanism enhancing plant reproductive success may not only include the association of floral traits with nectar taste but also the post-ingestive consequences of nectar consumption [84], their role in improving floral visitor health may also affect insect fidelity to specific flowers (as do other classes of nectar compounds; see the other paragraphs).

## 4. Nectar Alkaloids

Alkaloids are basic nitrogen compounds (mostly heterocyclic) [34] whose distribution among living organisms is limited [85]. Most alkaloids have basic properties, are bio-synthesised from amino acids, and show a wide variety of chemical structures. Extensive sampling of hundreds of plant species has demonstrated that they are common in the nectar of many plants [8,86,87].

Again, the occurrence and maintenance of potentially toxic alkaloids in floral nectar has been explained, like in the case of other NSCs, by stating that their presence may be beneficial to the plant by deterring less specialised floral visitors–which would presumably carry a smaller amount of co-specific pollen [8] or nectar thieves and/or robbers [37]. The study conducted by Barlow et al. [18] confirmed that nectar alkaloids in specialised *Aconitum* flowers deter thieving by bumblebees, although they may have co-evolved with specific patterns of nectar secretion aimed at maintaining the benefits of specialised plant-pollinator relationships. On the contrary, though, Haber et al. [88] found that most floral nectars containing alkaloids were willingly accepted and exploited by ants, indicating that they may not always be an effective barrier against theft of nectar and that their role may be more complex. For example, pyrrolizidine alkaloids have been suggested to represent an adaptation to exclude lepidopterans from exploiting the nectar of several plant families, although some specialised butterflies and moths seem attracted by these compounds [89], collecting volatile derivatives of the alkaloids and using them in predator defence and courtship [90,91].

Concentrations of nectar alkaloids that are sufficiently high to be a deterrent may also benefit plants by increasing their pollen export [92] or optimising the number of flower visitors per volume of nectar produced, allowing plants to reduce nectar production and energy investment [93].

Another possible ecological meaning attributed to alkaloids is again antibacterial or bacteriostatic and antifungal functions that limit microbial growth [11,15,50]. Curiously, the study by Fridman et al. [52] on the effects of certain nectar alkaloids did not confirm any effect in controlling bacterial growth. Nonetheless, insect pollinators could benefit from the intake of alkaloids. Alkaloids may play a prophylactic or therapeutic role by reducing the pathogen load of insects [94], and honeybees may actively search for alkaloid-enriched nectar to keep pathogens at bay [95].

What makes nectar alkaloids particularly intriguing are their neuroactive effects on floral visitors [12]. Many alkaloids are known to have strong biological activity, explained by their structural relationship with important neurotransmitters [96]. Alkaloids include good examples of compounds that may improve pollination services without benefiting floral visitors [11]. For instance, nicotine affects learning: at natural doses, bees learn the colour of flowers containing nicotine more efficiently than the colour of flowers offering the same nutritional value but without nicotine [26]. Even more interestingly, after experiencing flowers containing nicotine, bees become faithful to the flowers, even when the reward offered becomes suboptimal compared to other available food resources [26]. Similarly, Wright et al. [19] found enhanced memory of reward in bees that were fed solutions containing caffeine. This led them to postulate that memory enhancement can provide an evolutionary advantage to plants through the fidelity of free-flying bees to a caffeine-containing reward. Speculation on the enhancement of plant fitness was somehow confirmed by the subsequent essay of Thompson et al. [97] on artificial flowers: pollination by bumblebees was higher for flowers containing caffeine. Arnold et al. [98] also used robotic flowers to provide evidence that inexperienced bumblebees, primed in the nest with caffeine and a target odour, made more initial visits to flowers emitting the target odour than did control bees or those primed with odour alone. Caffeine-primed bees tended to more quickly improve their floral handling time. Although the effects of caffeine were short-lived, they showed that the food-locating behaviour of free-flying bumblebees can be enhanced by caffeine provided in the nest.

## 5. Nectar Non-Protein Amino Acids

Besides amino acids involved in building proteins, non-protein amino acids have also been found in nectar [14,35] and may account for up to 30–50% of nectar amino acid composition [14,99,100]. Non-protein amino acids are generally regarded as secondary metabolites because they are not directly involved in the primary metabolic pathways [101], although not all authors consider this classification appropriate [102]. Classification aside, many different functions have been attributed to nectar non-protein amino acids [14], particularly γ-amino butyric acid (GABA) and β-alanine, which are often the most frequent and abundant in floral nectar [14].

The ecological importance of nectar non-protein amino acids is now well established. As in the case of other secondary metabolites, an early ecological explanation for the presence of nectar non-protein amino acids was again the potential benefit gained by the plant by deterring scarcely efficient or inefficient pollinators ([14] and reference therein). Weakening this assumption, more recent findings show that these compounds do not alter nectar palatability [27] and have low toxicity [20,25].

Thus, more relevant roles of non-protein amino acids in floral nectar may be as neurotransmitters in insect nervous systems [12], muscle performance promoters [20,25], or feeding regulators of nectarivorous floral visitors [103,104]. A recent study conducted by Carlesso et al. [27] reported that honeybees were more likely to learn a scent when it signalled a sucrose solution containing β-alanine or GABA, suggesting that the latter enhances the learning of determined flower traits, thus favouring pollen transfer among conspecific individuals. Moreover, GABA proved to enhance memory retention. Some non-protein amino acids are suggested to reduce fatigue and sustain muscle activity in human beings [14,105,106,107]. Unfortunately, no study to investigate the effect of these compounds on the muscle activity of insects has been published. Nonetheless, taurine is found in the thoracic region of many insects and is associated with fully functional flight muscles [108], whilst the direct involvement of β-alanine in flight metabolism seems confirmed by Bogo et al. [20]: bumblebees fed with solutions enriched in β-alanine at natural concentrations showed the highest flying-index in a behavioural assay. Curiously, Felicioli et al. [25] reported that GABA-rather than β-alanine-enriched diets enhanced locomotion in *Osmia bicornis*.

GABA is known to stimulate taste chemoreceptors sensitive to sugars and increase feeding activity in caterpillars and adult beetles [16,24]. Indirect evidence of the phagostimulation activity of GABA comes from the finding that satiety in insects is opposed by simultaneous administration of GABA [103]. Nevertheless, it is speculated that the combination of GABA and NaCl, rather than GABA alone, plays a role in insect phagostimulation [104]. In fact, the absence of effects on the consumption of sucrose solution enriched with GABA alone in the forager honeybees tested by Carlesso et al. [27] stresses how studying the effects of different NSCs in isolation rather than their combined effects may yield a very different and unrealistic picture of how animal behaviour is influenced.

After all, this is just one of many examples where the effects of GABA coupled with other nectar chemicals help maintain the feeding rate of floral visitors [4]. GABA is also reported to be involved in plant communication with other organisms and accumulates in response to infection by fungi and bacteria [14,109,110].

## 6. Nectar Biogenic Amines

Biogenic amines are nitrogenous compounds known to function as neurotransmitters, neurohormones, and neuromodulators in invertebrates [111,112,113,114,115]. Thus, they shape behavioural patterns [116]. Their presence in floral nectar was recently reported for the first time in 15 species belonging to six plant orders [117]. Tyramine and octopamine are the two biogenic amines so far reported in floral nectar (Table 1). They are the invertebrate counterparts of vertebrate adrenergic transmitters that govern the so-called fight or flight response, namely quick adaptation to energy-demanding situations [28]. They are decarboxylation products of the amino acid tyrosine, and though tyramine is the biological precursor of octopamine, the two are considered to act as independent neurotransmitters [28]. The highest tyramine and octopamine concentrations so far (averaging about 0.07 mM) have been reported from the species *Citrus x meyeri* [117].

Tyramine has not only been found in nectar [117] but also in various foods of plant origin. This amine is associated with microbes with aminogenic activity in fermented foods and beverages [118], but little is known about why it is found, albeit in small amounts, in fruits, flowers, seeds, and other parts of plants [119].

Landete et al. [120] investigated the production of biogenic amines from selected strains of yeast, acidolactic bacteria, and acetic bacteria found in wine. Some of the yeast genera identified may also be found in the floral nectar of different plant species [121,122]. In any case, the ability to produce tyramine and other biogenic amines is correlated more with strain than species [123]. Yeasts do not appear to be the main producers of the amines found in wine, attributed to lactic-acid bacteria [120,124,125] that decarboxylate precursor amino acids, tyrosine in the case of tyramine and octopamine.

Besides being produced in nectar by microorganisms that decarboxylate amino acids, tyramine produced by endogenous enzymes such as tyrosine decarboxylase can also be naturally present in various parts of plants or their derivatives [126,127,128]. According to Servillo et al. [129], tyramine and its methylated forms, present in Citrus plants, are the products of specific pathways involved in response to attack by insects or other herbivores and pathogens, as they act as neurotransmitters that can modify various behaviours related to flight, feeding, and memory [130], and thus herbivore activity.

The enzyme tyrosine decarboxylase appears to be ubiquitous and implicated in various metabolic pathways where tyramine is the first product and in turn the precursor of many other molecules, including dopamine, octopamine, and a wide variety of alkaloids [131], implicated in defence against biotic and abiotic stressors [132]. The production of tyramine and other amines may be induced for the defence of the plant itself. Hydroxycinnamic acid amides, including tyramine-derived neutral amides, appear to be directly involved in plant defence against pathogens [133,134,135,136,137].

Since biogenic amines seem to have such important effects on the invertebrate nervous system, several studies have focused on insects, demonstrating that consumption of these substances modulates behavioural traits such as motivation [110], reward-seeking [21,138,139], learning [140,141,142] and social communication [21,31,130,143] (Table 1). Octopamine and tyramine both play an essential role in regulating basic motor functions. They differentially affect flight in honeybees when injected in the thorax, octopamine increasing flight and tyramine decreasing it [28,144].

Regarding the effects of biogenic amines on food-source communication and exploitation, Barron et al. [31] showed that octopamine increases the likelihood of dancing by honeybees, and Linn et al. [143] found that honeybees treated orally with octopamine were less likely to heed social information from waggle dances. This means that even if the food source bees find is poor, they are more likely to retain their personal information than to heed indications of a richer source. This evidence supports the hypothesis that nectar octopamine can increase bee faithfulness to a plant species and may favour its reproductive success. The results of Cnaani et al. [116] on bumblebees, seem to challenge this view. The authors showed that an octopamine-laden solution shortens the time bees need to change their visiting behaviour once they acquire information on changes in food source availability, making them able to direct their visits more promptly to better food sources in a scenario where the pattern of food availability is changing.

Besides being described as an enhancer of foraging activity [21,138,145], octopamine has also been demonstrated to be involved in the short-term regulation of forager behaviour in honeybee colonies, regulating the type of food source to which foragers direct their collection activity. Giray et al. [146] report that higher percentages of foragers treated with octopamine, but not those treated with tyramine, shifted from pollen-collection to nectar- or water-collection. Nectar-collecting bees treated orally with octopamine also showed a greater likelihood of switching their activity to the collection of water or nectar with lower sugar concentrations. Analysed from a plant perspective, both results suggest a trend directing bees to less valuable resources and may be explained by the effects of biogenic amines on perception. It is worth mentioning that some studies have provided evidence that both octopamine and tyramine enhance sucrose responsiveness [147,148,149]. This means that administration of both compounds lowers the sucrose response threshold, i.e., their consumption lowers the sucrose concentration necessary to elicit the proboscis extension reflex [148], enhancing bee perception of the value of a food source. It is worth highlighting, however, that in the above cases, concentrations of biogenic amines hundreds or even thousands of times greater than those occurring naturally in floral nectar were studied in isolation (Table 1). The study by Muth et al. [116] has the merit of providing the first insights into the effects of the administration of nectar-like concentrations of combinations of compounds on bee behaviour. Curiously, the authors found that tyramine and octopamine, given together, did not enhance sucrose responsiveness but instead seemed to erase the taste aversion for caffeine that bees showed when the alkaloid was tested alone. Similarly, the effect of caffeine on long-term memory was also erased by the co-administration of tyramine and octopamine, which did not exert any influence on their own.

**Table 1 plants-12-00550-t001:** Studies about the effects of tyramine (TA) and octopamine (OA) on bees. = the concentration used in the study is similar to that naturally occurring in nectar and reported for the first time by Muth et al. [117]; + the concentrations used in the study are higher by one order of magnitude for each +.

Reference	Model Species	Chemical	Method	Conc.	Effect
[140]	*Apis mellifera*	octopamine (serotonine and dopamine)	injection into the brain	0.05 mM =	OA enhanced responsiveness to olfactory stimuli
[141]	*Apis mellifera*	octopamine	injection into the brain	0.1 mM =	OA induced associative learning
[138]	*Apis mellifera*	octopamine and tyramine	oral ingestion	2 mg/mL +++	OA increased the number of new foragers, TA did not
[147]	*Apis mellifera*	Octopamine and tyramine (and dopamine)	oral ingestion, injection into the thorax	various concentrations, the lowest OA: 1 mM = TA: 0.01 mM =	At nectar-like concentrations, OA and TA didn’t affect sucrose responsiveness
[145]	*Apis mellifera*	octopamine	oral ingestion	2 mg/mL +++	OA increased responsiveness to brood pheromone, stimulating foraging
[116]	*Bombus impatiens*	octopamine	oral ingestion	various concentrations, the lowest at 2 mg/mL +++	OA shortened the time bees needed to direct their visits to a better food source
[148]	*Apis mellifera*	octopamine	Oral ingestion	various concentrations, the lowest at 20 µg/mL =	OA increased sucrose responsiveness (also at the nectar-like concentration)
[144]	*Apis mellifera*	octopamine and tyramine	injection into the haemolymph	various concentrations, the lowest at 0.05 mM =	OA and TA reduced walking and increased grooming and standing, with greater effects at higher concentration.
[146]	*Apis mellifera*	octopamine and tyramine	oral ingestion	various concentrations, the lowest at 125 µg/mL +	OA induced a switch in the type of collected material and affected sucrose responsiveness.
[31]	*Apis mellifera*	octopamine	oral ingestion	10.5 mM +++	OA increased the reporting of source value in dances.
[142]	*Apis mellifera*	octopamine (and dopamine)	oral ingestion	various concentrations, the lowest at 0.25 mg/mL ++	OA negatively influenced punishment learning.
[149]	*Melipona* *scutellaris*	octopamine	oral ingestion	various concentrations, the lowest at 10 mM ++	OA increased sucrose responsiveness.
[139]	*Apis mellifera*	octopamine	oral ingestion	10 mM +++	OA modified the probability that foragers switched the type of collected materia.
[21]	*Plebeia droryana*	octopamine	oral ingestion	10 mM +++	OA increased bee feeding and the frequency of individual foragin.
[143]	*Apis mellifera*	octopamine (and dopamine)	oral ingestion	2 mg/mL +++	Bees treated with OA followed fewer dances, increasing the use of private information.
[117]	*Bombus* *impatiens*	octopamine and tyramine (coupled)	oral ingestion	OA: 8 µg/mL * TA: 10 µg/mL *	OA + TA interacted with caffeine to alter key aspects of bee behavior.

* Concentrations within the range found in the nectar of *Citrus* × *meyeri*.

## 7. Intraspecific Variability of Nectar Secondary Compounds

The within-species variability of NSCs has rarely been investigated. The few studies highlight wide variability at the level of individual plants and patches within a population, as well as between populations [87,150,151]. Concerning cultivated plants, variability in NSCs has also been demonstrated between cultivars [87]. Although the qualitative composition of NSCs seems to overlap somewhat in different populations, quantitative composition differs by orders of magnitude [87,150]. Since the effects of NSCs are dose-dependent [11,19,50], this large quantitative variability makes it difficult to predict the effect that a specific compound may exert on a certain type of pollinator in a natural ecological context. It is precisely this high quantitative variability of nectar secondary compounds that may affect pollinator foraging behaviour. For example, nicotine concentration in the flower nectar of *Nicotiana attenuata*, unlike that found in other vegetative tissues, is known to vary unpredictably within and between populations, as well as between flowers of the inflorescence of the same individual [150]. This unpredictable variability of nicotine in floral nectar, particularly within an inflorescence, promotes outcrossing, probably because it keeps hummingbirds (the natural pollinators of this species) searching for low-nicotine flowers on a plant, enhancing their movement between flowers [17]. It appears clear that for the correct interpretation of the role of NSCs in determining the effects on plant reproductive fitness, the mating system of the species must be kept into consideration. However, the case of nectar nicotine allows a certain degree of generalization; this is because the compound is found in some self-compatible species of the genus *Nicotiana* whose reproductive output benefits from cross-pollination provided by animal visitors [152,153].

Nectar-dwelling microorganisms are a possible source of NSC variability. Several traits of the chemical environment of floral nectar, such as high sugar content, specific proteins [5], and specific secondary compounds (see previous sections) with known antimicrobial activity, impede the growth of most microorganisms. Nonetheless, specialised yeasts and bacteria that can cope with this “defence arsenal” are common inhabitants of floral nectar [51,154,155]. The presence and proliferation of these microorganisms drastically affect the chemical composition of nectar, generally lowering sugar and amino acid concentrations [53,156,157]. It is also demonstrated that nectar-dwelling microbes may alter levels of secondary compounds. Experiments using synthetic nectars spiked with secondary compounds and an array of inoculated microorganisms highlighted that the bacteria *Erwinia* sp. and *Gluconobacter* sp. and the yeast *Metschnikowia reukaufii* may reduce concentrations of nicotine and aucubin (an iridoid glycoside) [158]. Besides lowering the concentrations of nectar secondary compounds, it recently revived the interest–raised more than a century ago–in the idea that nectar-inhabiting microorganisms themselves can be a source of nectar secondary compounds not secreted by the plant. Biogenic amines, which were very recently detected in nectar [117], may be a class of compounds produced by microorganisms decarboxylating amino acids during the fermentation of nectar [50,118,120].

Since the main vectors transporting nectar-dwelling microorganisms from flower to flower are floral visitors [159,160,161], whose foraging activity is not homogeneous among all the flowers of a plant or of a population, nectar-dwelling microorganisms [159] and possible modifications in nectar chemistry [156] turn out to be spatially distributed, thus contributing to greater quantitative and possibly also qualitative variability of NSCs.

Another possible source of variability of secondary compounds in floral nectar is the activity of herbivores, which is obviously not homogeneous within or between populations. Leaf herbivory of *Nicotiana tabacum* by *Manduca sexta* increases alkaloid levels in floral nectar, indicating that interactions between species, involving leaf and floral tissues, are connected [162].

Besides biotic factors such as the above, abiotic drivers too may affect NSC concentrations. For example, nutrient abundance may affect concentrations of alkaloids in leaves and nectar [162].

## 8. Evolutionary Considerations on the Origin of Nectar Secondary Compounds

From the above, at least three other general functions can be recognised for nectar beyond food rewards for pollinators: (1) defence against microorganisms; (2) deterrence of exploiters (nectar thieves or robbers *sensu* [163]) and less efficient pollinators by changes in nectar palatability (pre-ingestive effects) or toxic effects; (3) modulation of insect mobility and behavioural traits (post-ingestive effects). Defence against microorganisms is common to all classes of NSCs [11,14,15,47,48,129,164,165]. Nectar first appeared in Palaeozoic fern clades [166], when few insects had yet evolved, defence against microorganisms may have been the original function of NSCs. In that era, nectar was not involved in plant interaction with insects. According to the “leaky phloem” hypothesis [167], nectaries were probably a kind of “sap valve” that exuded excess sugars. These sugary exudations may have been infected with microorganisms, some of which may have been pathogens exploiting nectarostomata to enter plant tissues [168]. Thus, plants needed protection against microbe proliferation. Regarding an alternative or concomitant hypothesis on the origin of NSCs, secondary compounds in nectar can be considered a pleiotropic trait, i.e., they occur in other plant organs (leaves, stems), protecting against herbivory, and are transported passively by phloem/xylem during nectar production [11,15]. The oldest plant–insect relationship is the predation of plants by herbivores. Plants underwent natural selection, evolving chemical defences based on secondary metabolites to cope with herbivory. The first arthropods and insects in the Silurian period may have been herbivorous, driving the selection of anti-herbivory secondary compounds in plant tissues, and these compounds presumably flowed passively into the nectar. Anti-herbivory functions are today recognised for all classes of NSCs (see previous sections). These molecules probably interacted with mutualistic insects, namely defenders and pollinators, when they evolved. Most “modern” mutualist insects (Diptera, Lepidoptera, and Hymenoptera including ants) radiated 125–90 Mya in the early-middle Cretaceous period, simultaneously with angiosperms [169]. They presumably drove plant selection towards optimal (low) concentrations of secondary metabolites in the secretions they fed on, while plants probably started to “manipulate” insect behaviour pharmacologically by secreting neuroactive compounds into nectar, thus improving their own fitness. In this regard, it is noteworthy that true nectar is lacking in gymnosperms, but their pollination drops can be considered an ecological analogue of angiosperm floral nectar [170]. Interestingly, β-alanine, a non-protein amino acid with neuroactive properties [12], was detected in the pollination drop of ambophilous gymnosperms (i.e., gnetophytes), in which pollination is performed by wind and insects feeding on pollination drops, but not in solely wind-pollinated species [171].

The presence of specific secondary compounds in nectar can also be explained from a microorganism perspective. Most recent hypotheses see nectar as an active interface between flowers and pollinators, in which microorganisms that colonise nectar also play an essential role [50]. These, through their metabolism, can affect nectar chemistry, modifying its olfactory attractiveness [53,172,173] and possibly synthesising secondary compounds or modifying the profile of existing ones, thus changing the behaviour of pollinators. Thus, the distribution of microorganisms in a population of flowers is ensured, using flower visitors as vectors [174]. In this case, evolution of the chemical profile of floral nectar and other floral traits [175] could be driven by the need of microorganisms to be transferred and to reproduce in other flowers.

It seems likely that multiple drivers, namely plant reproductive fitness, microorganism dispersal, and climatic and environmental parameters, were responsible for the evolution of the complex chemical profile of the modern floral nectar of angiosperms [176].

## 9. Future Research Perspectives

While many studies concern nectar volume and chemical composition in terms of sugars, and to a lesser extent amino acids, comparatively few studies concern the array of nectar secondary compounds [11]. Our knowledge of their distribution at a systematic level is therefore limited. Although Palmer-Young et al. [87] were the first to take a systematic non-targeted metabolomic approach to analysing secondary metabolites, their study only concerns 31 species. The determination of secondary compounds in different systematic contexts is therefore highly recommended for future research.

Another limitation of our knowledge of nectar secondary compounds is that their effects have only been studied in bees, with most of the focus on honeybees and bumblebees [19,20,26,27,98,117,177]. Future research, therefore, needs to consider other important taxa of insect pollinators such as flies, butterflies, and solitary bees.

More study is also needed on the link between nectar secondary compounds, pollination efficiency, and plant fitness in general. Although the “nectar manipulation” hypothesis postulates that NSCs are tools by which plants affect pollinator foraging behaviour, increasing plant reproductive output [33], we have little and inconsistent evidence of this relationship. In *Nicotiana attenuata,* both attractant (benzyl acetone) and repellent (nicotine) compounds are required to maximise pollen export (male function), capsule and seed siring (female function) and flower visitation by native pollinators, whereas nicotine has been reported to reduce florivory and nectar theft and/or robbery [17,93]. High levels of nectar alkaloids may benefit plants of *Gelsemium sempervirens* via the increased male function (pollen export) under a limited set of ecological conditions (abundant efficient pollinators, large floral displays) but have no effect on female function (seed production) [92]. More recent papers dealing with the effect of specific nectar compounds on pollinator behaviour ignore or only partly investigate the possible outcomes for plant reproduction [19,20,25,26,27,117]. Using artificial flowers, it has been demonstrated that caffeine-laced nectar brings more visits by bumblebees and more pollen analogue (dye particles) than nectar without caffeine [97].

The lack of clear evidence of links between NSCs, pollinator behaviour, and plant reproductive output is important since such links are pivotal for considering NSCs to be adaptive and therefore subject to selection. In the absence of data for many species, we cannot exclude the possibility that nectar secondary compounds are non-adaptive and just a pleiotropic trait (see before) [11,15]. It should in any case be highlighted that the identity and concentration of specific secondary compounds may vary between nectar and leaves, suggesting that the production or allocation of secondary compounds may be independently regulated in each plant part [178], in turn indicating possible selection pressure by different drivers. The presence of secondary compounds in nectar is probably the result of adaptive and non-adaptive factors, as suggested by Manson et al. [178].

One more point that needs further attention is the possible interactive effect exerted by a mix of NSCs. In most cases, the effects of NCSs on insects have been studied experimentally in isolation [19,20,25,26,27,97]. This is different from the natural ecological context where nectar-feeding insects experience a complex phytochemical nectar environment characterised by a mixture of substances. Very recent papers underline the importance of studying the effect of mixtures of NSCs and of finding interactive effects with pairs of compounds. Muth et al. [117] revealed that a combination of tyramine and octopamine, in a range of concentrations occurring naturally in nectar, had no effect on insect behaviour, whereas when combined with caffeine, they alter key traits of bumblebee (*Bombus impatiens*) behaviour, such as sucrose responsiveness, long-term memory, and floral preferences. Artificial feeding experiments by Marchi et al. [177] found that single compounds such as arginine and caffeine increased honeybee learning performance, but that insect memory retention only increased significantly when feeding treatments offered a combination of the two compounds. These findings highlight that studying the effects of NSCs as single molecules is too simplistic and that it is necessary to test mixtures of NSCs, at concentrations occurring naturally in nectar, also combined with other substances.

A further element of complexity is that nectar chemistry (including NSCs) may affect pollinator behaviour through other floral traits such as colour. For example, bumblebees (*Bombus impatiens*) that had experience with blue flowers preferred blue regardless of nectar chemistry. In contrast, bees having prior experience with white flowers only preferred white in the case of control treatment, whereas bees exposed to caffeine and ethanol showed no preference [179].

Another aspect that needs to be considered is the possibility that certain contaminants may alter the effects of NSCs on insect behaviour and other traits. It was demonstrated that the common neonicotinoid imidacloprid attenuated the positive effects of certain NSCs, while an NSC-enriched diet increased the negative effects of pesticide exposure [180].

Finally, a further consideration worthy of attention is the link between certain classes of NSCs and abiotic stress. Since plants can synthesise a variety of secondary metabolites to cope with stress, levels of these substances are related to environmental changes. GABA, for example, is involved in drought and heat stress resistance in plants [181,182]. Higher temperatures, drought, and heat waves are expected to increase significantly in the near future in certain regions of our planet, according to the current climate change scenario [183]. An increase in GABA concentrations is likely in plant tissues to counteract increased stress. If this increase also spills over into the nectar, due to the general correlation between levels of secondary compounds in leaves and nectar [162], then plant-pollinator interactions could change, since the effects of GABA on bees are concentration-dependent [20].

## 10. Concluding Remarks

Today the ecological functions of nectar are recognised to be far more than a simple food reward for pollinators [184]. The complex chemical composition of floral nectar, especially in terms of primary and above all secondary compounds, reflects additional functions that make nectar a plant interface for complex, multi-faceted biotic interactions involving plants, pollinators, nectar exploiters, and nectar-dwelling microorganisms [184,185] (Figure 2). Although the “nectar manipulation” hypothesis [33] still has gaps, it is a good framework for shaping future studies in the field of nectar ecology and evolution, also considering the expected scenarios of climate change. In any case, the manipulation of behavioural traits of pollinators is just one facet of the multi-faceted interactions mediated by floral nectar, which should therefore be considered from a more comprehensive perspective. The role of microorganisms, yeasts, and bacteria in these multifaceted interactions seems largely overlooked [174,186,187], limiting an overall understanding of their role in pollinator behaviour, plant-pollinator interactions, and plant fitness.

## Figures and Tables

**Figure 1 plants-12-00550-f001:**
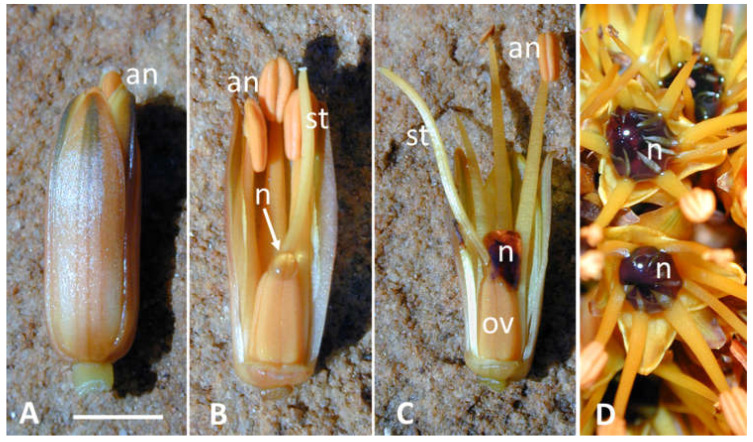
Coloured nectar of *Aloe castanea*. (**A**,**B**) young opening flower containing a small drop of uncoloured nectar (arrow). (**C**,**D**) older flower(s) (2–4 h after opening) with dark-red coloured nectar. an = anther; n = nectar; ov = ovary; st = style. Bar = 5 mm.

**Figure 2 plants-12-00550-f002:**
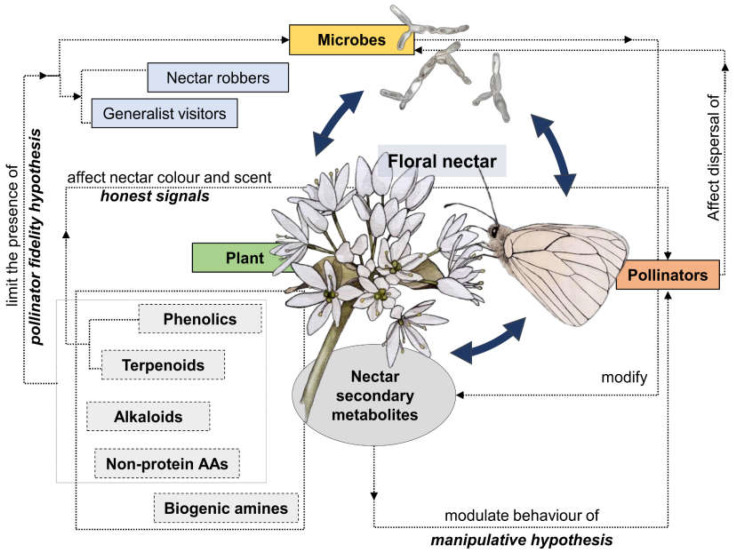
Network of nectar-mediated complex relationships involving plants, microbes, and pollinators. Nectar secondary compounds are pivotal in shaping such interactions.

## Data Availability

Not applicable.

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
