# Peer review of "Secondary Metabolites in Nectar-Mediated Plant-Pollinator Relationships"

_plants, 2023, doi:10.3390/plants12030550_

Round 1

Reviewer 1 Report

I have completed the review of the manuscript. The present study addresses the chemical role of nectar secondary compounds (NSCs) in the ecological context with a focus on non-protein amino acids and biogenic amines. After considering the implications of the different classes of NSCs in the pollination scenario, were discussed the hypotheses regarding the evolution of such complex nectar profiles and providing insights for future research on plant-pollinator relationships.

Overall, the manuscript is well structured and written. The main aims of the study on the functions of NSCs, phenols, terpenoids, alkaloids, nonprotein amino acids and biogenic amines is well discussed in depth regarding evolutionary ecological aspects.

The main contribution of the manuscript is to compile recent knowledge on the function of secondary metabolites in nectar and their ecological implications. The study also serves as a guide for future studies of floral nectar chemistry and plant-pollinator interactions. 

“Introduction”

The introduction is well written and looks fine to me.

The perspective and conclusions are well supported based in current literature.

Author Response

Thanks to the reviewer for this positive comment about our manuscript.

Reviewer 2 Report

I chose to review this MS because although I have taught chemical ecology and coevolution for many years, I knew relatively little about secondary compounds in nectar and their potential ecological roles. I had many questions, especially regarding potential nectar toxicity (the famous case of Aesculus californica and honeybees (admittedly, non-native)  comes to mind) and was delighted to find literally all the questions I had jotted down prior to this review answered beautifully in the paper--you have a very appreciative reader in me and I suspect many others will feel similarly. 

Author Response

Many thanks to the reviewer for this very positive comment about the manuscript.

Reviewer 3 Report

Review L. S-G

Secondary Metabolites in Nectar-Mediated Plant-Pollinator Relationships

Authors: Marta Barberis, Daniele Calabrese, Marta Galloni, Massimo Nepi * 

Journal: Plants
Manuscript ID: plants-2146292
Type of manuscript: Review

This a comprehensive review of the floral chemistry of secondary compounds found in nectar which mediate plant-pollinator relationships with the goal of summarizing current knowledge in the field. The different functions of NSCs are discussed. In addition,the different evolutionary theories on the origins of NSCs are summarized.

Review: This is a well written, high quality, thoroughly referenced review of an important topic to pollination ecologists, chemical ecologists, evolutionary ecologists interested in plant-insect-microbe interactions.  

The review uses both recent and foundational references.

Specific comments 

 Line 499-503 This is an excellent and important point with an example discussed earlier in the manuscript starting on Line 265.  In nature, the phytochemical nectar environment is complex.  Testing one compound at a time may not reveal pollinator-relevant interactions. Perhaps move this up in the manuscript.

This is just a suggestion: My comment is that the chemical ecology research on this topic is very heavily focused on Apis mellifera, with some research on Bombus species. Most of the pollinating bees on Earth are solitary bees, not social bees, I think this represents a significant gap in knowledge due to their behavioral, nesting, ecological and other differences which may affect their plant relationships.

Scope and quality: Yes, this manuscript fits the journals scope and is of a high quality.

Significance: This is a significant contribution to the topic. The conclusions are balanced and supported by the references cited but of course more research needs to be done on this topic.

Author Response

Reviewer's comment

Line 499-503: This is an excellent and important point with an example discussed earlier in the manuscript starting on Line 265. In nature, the phytochemical nectar environment is complex. Testing one compound at a time may not reveal pollinator-relevant interactions. Perhaps move this up in the manuscript.

Our response

We thank the reviewer for the suggestion. We decided to mention this aspect earlier in the manuscript (as suggested), but to keep also the same point in the “Future research perspectives” paragraph, where originally that was mentioned. In fact, we agree that it’s worth mentioning this very important point earlier in the paper, coupled with the practical example of GABA and NaCl, but we also consider this point as one that future research should consider. The sentence added near the example of GABA and NaCl is: “In fact, the absence of effects on the consumption of sucrose solution enriched with GABA alone in the forager honeybees tested by Carlesso et al. [20] stresses how studying the effects of different NSCs in isolation rather than their effects when combined together may yield a very different and unrealistic picture of how animal behaviour is influenced”.

Reviewer's comment

This is just a suggestion: my comment is that the chemical ecology research on this topic is very heavily focused on Apis mellifera, with some research on Bombus species. Most of the pollinating bees on Earth are solitary bees, not social bees, I think this represents a significant gap in knowledge due to their behavioral, nesting, ecological and other differences which may affect their plant relationships.

Our response

We agree and thank the reviewer for pointing out this crucial aspect. We mentioned this point in the “Future research perspectives”, with the following sentence: “Another limitation of our knowledge of nectar secondary compounds is that their effects have only been studied in bees, with most emphasis on honeybees and bumblebees [13,15,19-21,96,169]. Future research therefore needs to consider other important taxa of insect pollinators such as flies, butterflies and solitary bees”.

Reviewer 4 Report

Dear authors,

I put all my comments directly into the manuscript.

Author Response

Our point-by-point response to the reviewer comments is reported in the attached file. The reviewer's comments are in black, our response is in blue.

Round 2

Reviewer 4 Report

It seems that authors answered the questions raised during the review process properly and significantly improved the manuscript.